# Pain, fatigue, and associated gene expressions over chemotherapy in patients with colorectal cancer

Weizi Wu[1,2], Aolan Li[3], Vijender Singh[4], Andrew Salner[5], Ming-Hui Chen[3], Michelle P. Judge[1], Xiaomei Cong[2], Wanli Xu[1]*

1 School of Nursing, University of Connecticut, Storrs, Connecticut, United States of America, 2 Yale University School of Nursing, Orange, Connecticut, United States of America, 3 Department of Statistics, University of Connecticut, Storrs, Connecticut, United States of America, 4 Computational Biology Core, University of Connecticut Institute for Systems Genomics, Storrs, Connecticut, United States of America, 5 Hartford HealthCare Cancer Institute, Hartford, Connecticut, United States of America

* wanli.xu@uconn.edu

## Abstract

### Context

Patients with colorectal cancer undergoing chemotherapy often experience significant pain and fatigue. Limitations in understanding the complex phenotypes and biological mechanisms of these symptoms hinder effective interventions.

### Objectives

This study aimed to identify the pain and fatigue patterns during one chemotherapy cycle and associated gene expression profiles.

### Method

In a prospective longitudinal study, 34 patients with colorectal cancer from a major cancer center in the Northeastern US were recruited. Self-reported outcome measures of pain and fatigue and blood samples were collected at baseline, post-chemotherapy, and at the end of the chemotherapy cycle. RNA sequencing followed by differential expression analysis identified changes in gene expression. Linear mixed models examined associations between symptoms and possible biomarkers over time.

### Results

The sample had a mean age of 58.4 years old, with 97% being white and non-Hispanic. Among participants, 44.1% had stage III cancer, and 26.5% were undergoing initial chemotherapy. Abdominal pain was the most frequently reported symptom. Fatigue levels significantly worsened post-chemotherapy (P=0.011) and after recovery (P=0.018). Critical pathways involved inflammatory response and myeloid cell

**Data availability statement:** All relevant data are within the paper and its Supporting Information files.

**Funding:** This work was made possible by the Oncology Nursing Foundation, which provided research support through the RE01 grant in 2019. No specific grant number was assigned. Wanli Xu, the corresponding author, received support for this research. Weizi Wu, the first author, received a Sigma Theta Tau International (STTI) Mu Chapter Research Grant to support this work. No specific grant number was assigned. There was no additional external funding received for this study.

**Competing interests:** The authors have no relevant financial or non-financial interests to disclose.

development (FDR < 5%). Mixed-effect linear regression analysis revealed statistically significant associations between the upregulation of LILRA6 and higher pain interference (β = −6.621, p = 0.010) and fatigue (β = −6.621, p = 0.010), as well as between the downregulation of CACNG6 (β = −1.043, p = 0.047) and PRSS33 upregulation (β = 1.384, p = 0.038) and increased pain interference. Given the small sample size, these findings should be interpreted with caution.

## Conclusion

These findings suggest inflammation and specific biomarkers may drive pain and fatigue during chemotherapy. Further preclinical models or clinical cohorts are needed to validate these results and explore potential implications for targeted interventions to reduce symptom burden in patients with colorectal cancer.

## Introduction

Colorectal cancer (CRC) is the second most common cancer in the United States, with an estimated 153,020 new cases and 52,550 estimated deaths in 2023 [1], imposing significant burdens on individuals and society [2]. While early screening and advances in treatment have greatly improved survival rates [3], chemotherapy remains a first-line treatment for stages II-IV CRC [4]. Cytotoxic effects, while aimed at damaging cancer cells' DNA or replication [5], also affect healthy cells' proliferation [6], leading to a range of adverse side effects. Patients undergoing chemotherapy often experience distressing and persistent symptoms that significantly impact their treatment adherence and quality of survivorship [7–10]. Pain and fatigue were among the most debilitating symptoms, affecting up to 70% of patients [7,10–15].

A significant challenge in developing effective interventions for chemotherapy-related pain and fatigue is the lack of understanding of the complexity of these symptoms and mechanisms that drive the interpersonal variabilities and intrapersonal changes of symptom phenotypes [16]. Emerging evidence points to the role of immune-inflammatory perturbations as a critical driver of chemotherapy-induced pain and fatigue [17–21]. Recent studies have shown that dysregulated mRNAs and long non-coding RNAs are primarily enriched in inflammatory and immune processes in the spinal cords of bone cancer models with pain [22–24]. In CRC animal models, nociceptive abdominal pain has been linked with inflammatory conditions, such as mucositis, as a result of chemotherapy-induced molecular changes in the tight junctions of gastrointestinal epithelial cells [17,18]. Similarly, in clinical practice, the use of oxaliplatin, a first-line chemotherapeutic agent for CRC, is associated with the development of peripheral neuropathy in up to 60% of patients [25]. This side effect is mainly due to damage to dorsal root ganglia neurons and neuroinflammation, which result in cold-sensitive sensory symptoms and neuropathic pain in the limbs [25,26]. Similar neuroinflammatory pathways have been identified in breast cancer survivors experiencing paclitaxel-induced peripheral neuropathy [27].

The complexity of fatigue mechanisms is similarly demonstrated by research showing that distinct inflammatory pathways may drive morning and evening fatigue in patients undergoing chemotherapy [28]. In a sample of 89 Hispanic/Latino patients with CRC, elevated fatigue levels were linked with the upregulation of B lymphocytes and CD8-positive T lymphocytes, alongside increased transcription factors involved in immune activation, such as nuclear factor κB (NF-κB), signal transducer, and activator of transcription (STAT) [21].

Existing mechanistic research remains limited to animal models, and few studies have explored gene expression dynamic changes triggered by chemotherapy that might contribute to pain and fatigue in patients with CRC [29–31]. RNA sequencing (RNA-seq), a powerful tool for transcriptome-wide analysis, offers a comprehensive and detailed gene expression regulation, holding the potential for identifying biomarkers and biological mechanisms underlying pain and fatigue. Therefore, the aims of this study are 1) to determine pain and fatigue trajectories and gene expression profiles throughout the chemotherapy cycle and 2) to explore pain- and fatigue-related differentially expressed genes and regulated biological processes.

## Methods

### Study design and setting

A prospective longitudinal study recruitment was conducted from December 19, 2019, to March 28, 2022, at a renowned cancer institute in the Northeastern US. Patients with CRC (n = 34) undergoing chemotherapy were recruited and followed up for one cycle (21 days for the CAPEOX regimen and 14 days for the FOLFOX regimen). Questionnaire data and blood samples were collected at three time points: 1-2 days before the chemotherapy cycle (visit 1), within two days after chemotherapy administration (visit 2), and at the end of one chemotherapy cycle (visit 3). We used the STROBE cohort checklist when writing our report [32].

### Samples

Patients included in this study were adults who were: 1). aged 18 or older, 2). diagnosed with clinical or pathological stage II, III, or IV CRC, 3). underwent the chemotherapy regimens, 4). voluntary to provide blood samples. Potential patients were excluded from the study if they were diagnosed with stage I CRC, were not receiving chemotherapy, or had a life expectancy of less than six months.

### Participant recruitment

The study protocol received approval from the Institutional Review Boards of the cancer center and the research institute. Healthcare providers' referral was the primary source of recruitment. Physicians helped identify potential patients and approached them during office visits to gauge their interest in the study. A study handout containing information about the study and the contact information for the research-designated phone line was provided. If patients expressed interest in participating in the study, research assistants followed all informed consent procedures. Upon written consent, patients were invited to in-person study visits.

### Survey questionnaire collection

Patients' demographics and medical history were extracted directly from clinical chart review, including age, income, employment, education, race, ethnicity, gender, clinical diagnosis (stage, grade, and comorbidities), and chemotherapy regimen. Questionnaires were collected using REDCap to measure pain and fatigue.

**Brief Pain Inventory Short Form (BPI-SF)**, a 13-item pain assessment tool [33], was used to measure pain severity and interference. The BPI-SF can sensitively assess the severity of pain, the most painful area, and the impact of pain on daily functions, as well as the change in pain relief in the past 24 hours and the past week on a 0-10 (*0 = no pain or*

*interference, 10 = the worst possible pain or complete interference*); A higher mean score indicating greater severity or interference. This tool is used worldwide with high internal consistency (0.81 < α < 0.95) and good construct validity [33,34].

**Functional Assessment of Chronic Illness Therapy – Fatigue (FACIT-F)** [35] was used to assess the fatigue level over the past week, 13 items with a 5-point Likert-type response *(0 = not at all to 4 = extremely)*. The total score is negatively associated with the level of fatigue as items are reversely summed to calculate the total score. FACIT-F has excellent reported reliability ranging from 0.82 to 0.91 and significant concurrent validity between the FACIT-F and Multidimensional Assessment of Fatigue scale scores [35].

## Biospecimen collection and RNA sequencing

**Blood samples** were collected using PAXgene® Blood RNA tubes at each time point. Sterile techniques were strictly followed for venous blood draws, and samples were collected by trained personnel. The PAXgene® samples were transported to the institute's Biobehavioral Lab and stored at −80°C after being kept at the normal room temperature for 2–48 hours. Blood samples were transported to the institute's Genome Innovation Center for RNA sequencing assay, where paired-end reads were obtained and stored in Xanadu Cluster, a secure platform hosted by the study institute's Computational Biology Core.

**RNA sequencing** is a high-throughput sequencing technology that identifies and quantifies RNA in a biological sample. This technique can detect gene fusions, mutations/SNPs, and changes in gene expression [36]. RNA extraction was performed first to isolate total RNA from blood samples, using a phenol-chloroform extraction to ensure high RNA purity and integrity. This procedure is followed by DNase treatment to remove any genomic DNA contamination. The main RNA seq includes library preparation and sequencing. Following the manufacturer's protocol (Illumina, San Diego, CA, USA), the process of preparing a complementary DNA (cDNA) library involves 1) RNA selection. Total RNA was quantified using the Illumina TruSeq Stranded mRNA Sample Preparation kit, and purity ratios were determined for each sample using the NanoDrop 2000 spectrophotometer and Agilent TapeStation 4200. 2) cDNA synthesis: RNA was reverse transcribed to cDNA because DNA is more stable, which allows for amplification (which uses DNA polymerases) and leverages more mature DNA sequencing technology. Illumina Transcriptome sequencing used Illumina NextSeq 500/550 sequencing by denaturing and diluting the libraries. Target read depth was achieved per sample with paired-end 75 bp reads.

## Data analysis

The data analysis in this study was performed using R statistic packages (Version 4.2.2). Descriptive analysis was used for demographic and clinical data. Distribution, outliers, and missing data were technically processed. Box plots were used to detect potential outliers. Missing data was imputed by multiple imputations using linear regressions.

**Gene differential expression throughout the chemotherapy cycle.** The RNA sequencing differential expression (DE) analysis comprised four main stages: preprocessing, mapping, post-processing, and data analysis (S1 Fig 1). During preprocessing, the data underwent quality assessment pre- and post-trimming using *fastqc* and *multiqc,* as well as trimming using *fastp*. In the mapping stage, we selected the human genome hg19 as the mapping index and aligned the reads to the reference genome with *HISTA*. For post-processing, the program *HTseq-count* was used to count the RNA fragments (i.e., read pairs) mapped to each annotated gene in the genome. Once the counts were generated, we applied R package *edgeR* to perform gene DE analysis [37]. A blocking model was performed to assess DE over a single chemotherapy cycle. Additionally, addictive models were conducted to make pairwise comparisons between visits, specifically comparing visit 1 to visit 2, visit 2 to visit 3, and visit 1 to visit 3. The false discovery rate (FDR) approach was performed to correct the P-value from multiple comparisons. Enrichment analysis was performed using the *GOseq and KEGG* packages. Mean-difference (MD) plots and enrichment dot plots illustrated the results for visualization by *clusterProfiler* package.

**Pain/fatigue trajectories related to differentially expressed genes.** Symptom trajectories were illustrated by *ggplot 2*. Linear mixed-effects models (LMMs) assessed associations between symptoms and differentially expressed genes, where pain severity, pain interference, and fatigue were outcomes and genes were independent variables. LMMs

are essential for analyzing longitudinal data with repeated measures because they can account for both fixed effects (representing population-level trends) and random effects (accounting for subject-specific variations). Chemotherapy regimens and cycles were considered as potential confounding factors. The gene selection for pain and fatigue models involves two key stages. Firstly, the potential gene pools were created for pain and fatigue models, identifying the most significant genes from the above DE analysis results (FDR < 0.05, |logFC| > 1.5). The identified gene's raw counts were scaled (log()) and standardized before further processing. Secondly, to determine the most relevant genes, the glmmLasso package was utilized, which provides a variable selection approach for generalized linear mixed models using L1-penalized estimation. The same pairwise comparisons were conducted to compare the differential expressions of distinct trajectories.

## Results

### Demographical and clinical characteristics

Thirty-four patients were included in this data analysis (Table 1). The average age was 58.2 ± 12.4 years old. The majority were white (97.0%), non-Hispanic (97.1%), capable of self-care (76.5%), overweight or obese (61.8%), married (67.6%), employed (61.8%), and earned more than $50,000 per year (81.8%). Slightly more than half were male (58.8%) and had a bachelor's or higher education background (52.9%).

Regarding diagnosis, 44.1% of the patients were diagnosed with stage III cancer, and 35.3% were diagnosed with stage IV cancer. Of these participants, 26.5% underwent their initial chemotherapy cycle at enrollment. 38.2% were treated with the FOLFOX regimen (5-Fluorouracil (5-FU), leucovorin, and oxaliplatin), and 35.3% of the participants received the CAPEOX regimen (capecitabine and oxaliplatin). Regarding the cancer site, 73.5% of the participants had colon cancer, while 23.5% had rectal cancer. Blood tests on white blood cells, hemoglobin, hematocrits, and platelets are within normal range over time.

### Pain and fatigue trajectory during one chemotherapy cycle

The findings presented dynamic fluctuations in pain and fatigue levels throughout the chemotherapy cycle (Fig 1). While the Repeated Measures ANOVA and post-hoc analyses did not indicate statistically significant differences in pain outcomes, significant changes were observed in post-hoc analysis of fatigue levels between visit 2 and both visit 1 (P = 0.011) and visit 3 (P = 0.018) (Table 2). Moreover, the total fatigue scores were lower than those reported in the U.S. general population samples (43.6 ± 9.4) [38] and cancer patient samples (36.9 ± 11.4) [39], indicating higher fatigue levels in this study samples. The abdomen was the most reported pain site, with incidence rates of 36.36%, 41.67%, and 37.68% across the three visits. The most frequently reported pain characteristics were aching (25%) and cramping (20%) following chemotherapy administration.

### Gene differential expression and visualization

Eighty-four blood samples were included for RNA sequencing. After filtering out low-quality reads, the average sequence length ranged from 130 to 144 bp. The overall mapping rate to the reference genome exceeded 88.6%, and 66,027 genes were retained for further differential gene expression analysis. Mean Difference (MD) plots (Fig 2) revealed significant alterations in gene expression at visit 2 compared to visits 1 and 3, while no significant differences were observed between visit 1 and visit 3. All differentially expressed genes with logFC greater than 1.5 (S1 Table 1). Enrichment analysis indicated symmetrical changes in biological processes across the three-time points (Fig 3). Notably, immune-inflammatory response pathways, including responses to bacterium and cytokine-mediated signaling, were significantly upregulated at visit 2 and downregulated at visit 3, with an FDR of less than 5%. In contrast, myeloid and erythrocyte-related pathways were downregulated at visit 2 and subsequently upregulated at visit 3.

**Table 1. Demographic and Clinical characteristics among CRC patients (N = 34).**

|  | Median | Mean | SD |
|---|---|---|---|
| Age | 58.5 | 58.2 | 12.4 |
|  | **Levels** | **Frequency (n)** | **Proportion (%)** |
| Sex | Male | 20 | 58.8 |
|  | Female | 14 | 41.2 |
| BMI | Normal | 13 | 38.2 |
|  | Obesity | 13 | 38.2 |
|  | Overweight | 8 | 23.6 |
| Race | White | 32 | 97.0 |
|  | Asian | 1 | 3.0 |
| Ethnicity | Non-Hispanic | 33 | 97.1 |
|  | Hispanic | 1 | 2.9 |
| Marital status | Married | 23 | 67.6 |
|  | Other status | 11 | 32.4 |
| Employment | Employed | 21 | 61.8 |
|  | Unemployed | 3 | 8.8 |
|  | Retired | 10 | 29.4 |
| Education | College or above | 18 | 52.9 |
|  | High School or GED | 5 | 14.7 |
|  | Some college | 11 | 32.4 |
| Annual income | <$50,000 | 6 | 18.2 |
|  | $50,000 - $99,999 | 17 | 51.5 |
|  | = or>$100,000 | 10 | 30.3 |
| Primary caregiver | Self-care | 26 | 76.5 |
|  | Spouse or partner | 8 | 23.5 |
| Chemotherapy cycle | < 2 | 9 | 26.5 |
|  | = or > 2 | 25 | 73.5 |
| Chemotherapy regimen | FOLFOX | 13 | 38.2 |
|  | CAPEOX | 12 | 35.3 |
|  | Others | 9 | 26.5 |
| Cancer Site | Colon | 25 | 73.5 |
|  | Rectal | 8 | 26.5 |
| Cancer stage | Stage II | 7 | 20.6 |
|  | Stage III | 15 | 44.1 |
|  | Stage IV | 12 | 35.3 |

\* Chemotherapy regimen FOLFOX: Leucovorin-Fluorouracil-Oxaliplatin; CAPEOX: capecitabine-oxaliplatin.

## Symptom trajectory and enriched biological pathways

To further explore the pain- and fatigue-related biological processes during chemotherapy, patients were categorized into two trajectories based on their symptom progression patterns over time. Given the sample size, we used a binary classification strategy to distinguish symptom trajectories. Individuals who reported increased pain scores at visit 2 compared to visit 1 were classified as belonging to Pain Trajectory 1, while those who reported unchanged or lowered pain scores were categorized as Pain Trajectory 2 (S1 Fig 2). Similar methods were applied to Fatigue Trajectory 1 and 2 (S1 Fig 2). No differences were observed when comparing the scores from visit 1 and visit 3 (Figs 4 and 6). Pain Trajectory 1 (pain

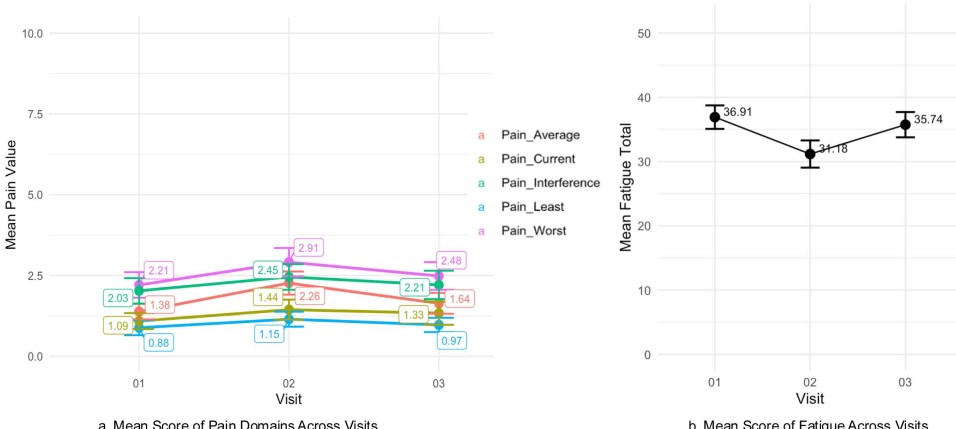

Fig 1. **Mean of pain and fatigue score over time.**

Table 2. **Repeated Measure ANOVA results for main outcomes (N = 34).**

| Outcomes | RM ANOVA | | | Post-hoc corrected P value [a] | | |
|---|---|---|---|---|---|---|
| | df | F | P | V2 vs V1 | V3 vs V2 | V3 vs V1 |
| Pain severity | 2 | 0.053 | 0.949 | 0.38 | 0.59 | 1 |
| Pain interference | 2 | 0.053 | 0.949 | 0.87 | – | 1 |
| Pain worst | 2 | 0.044 | 0.957 | 0.58 | 0.79 | 1 |
| Pain least | 2 | 0.016 | 0.984 | 1 | 1 | 1 |
| Pain current | 2 | 0.16 | 0.852 | 1 | 1 | 1 |
| Pain average | 2 | 0.51 | 0.602 | 0.11 | 0.12 | 0.37 |
| Fatigue total score | 2 | 1.269 | 0.286 | 0.012* | 0.018* | 0.831 |

[a]Post-hoc pairwise comparison with Bonferroni correction

*Corrected P value <0.05

V1: visit 1; V2: visit 2; V3: visit 3. Repeated Measure One-way ANOVA

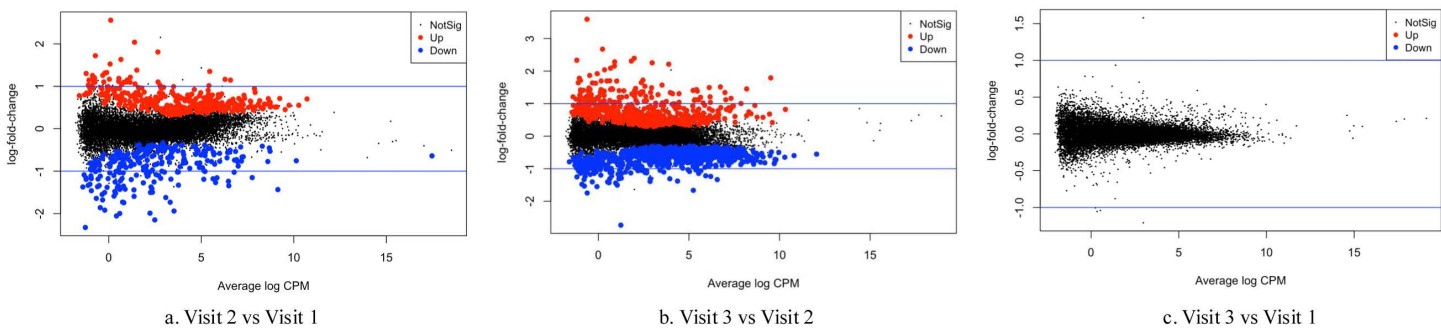

Fig 2. **Dynamic alterations in gene expression across study visits.** The mean difference (MD) plot was generated using the edgeR package (FDR < 0.05). Each data point denotes a gene, presenting the average log counts per million (CPM) on the x-axis, while the logs of fold change (logFC) are represented on the y-axis, depicting comparisons between Visit 2 versus Visit 1 (left panel), Visit 2 versus Visit 3 (middle panel), and Visit 3 versus Visit 1 (right panel). The red markers indicate significantly upregulated genes, the blue markers signify downregulated genes, and the black markers represent genes that do not exhibit significant differential expression.

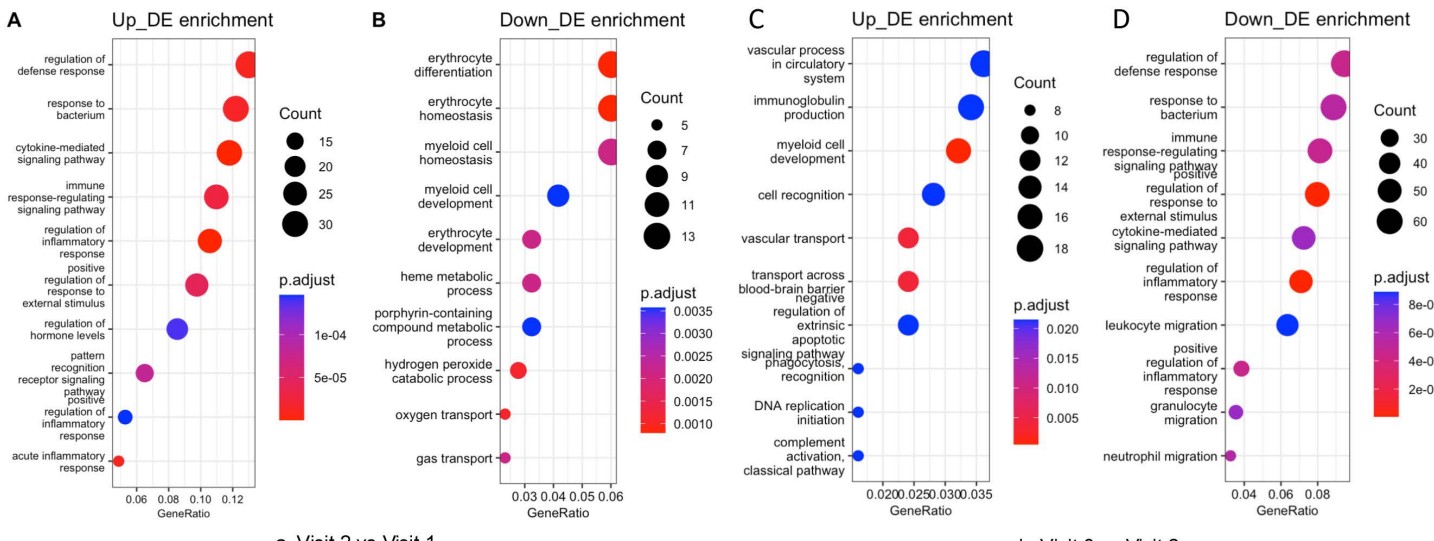

a. Visit 2 vs Visit 1                    b. Visit 3 vs Visit 2

**Fig 3. The bidirectional regulation of biological processes throughout the chemotherapy cycle.** The enrichment analysis of Gene Ontology (GO) terms reveals reciprocal activation patterns of pathways between Visit 2 and Visit 3. A (Up at V2 vs V1): The ten most significantly upregulated pathways at Visit 2 include, for instance, immune-inflammatory responses. B (Down at V2 vs V1): The ten most significantly downregulated pathways at Visit 2 encompass erythrocyte development. C (Up at V3 vs V2): The recovery of myeloid and erythrocyte pathways is observed during Visit 3. D (Down at V3 vs V2): Notable suppression of immune-inflammatory pathways occurs at Visit 3. The size of the bubbles corresponds to the number of genes associated with each pathway, while the color gradient indicates the adjusted p-value (FDR < 0.05).

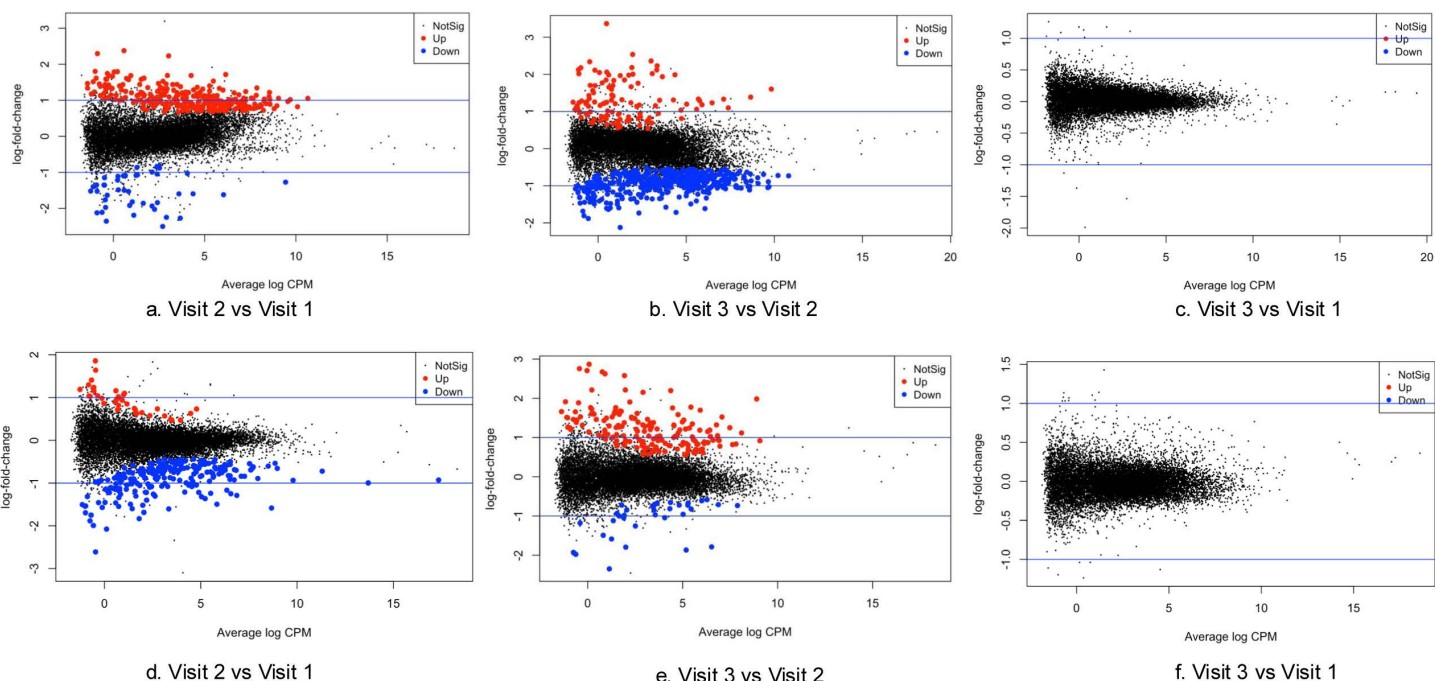

**Fig 4. Mean difference (MD) plot in pain trajectory groups over time.** The red dots indicate significantly upregulated genes, while the blue dots indicate downregulated genes. (Figs 4a, 4b, and 4c) presented the gene differential expression of pain trajectory 1 over time. Figs 4d, 4e, and 4f presented the gene differential expression of pain trajectory 2 over time.

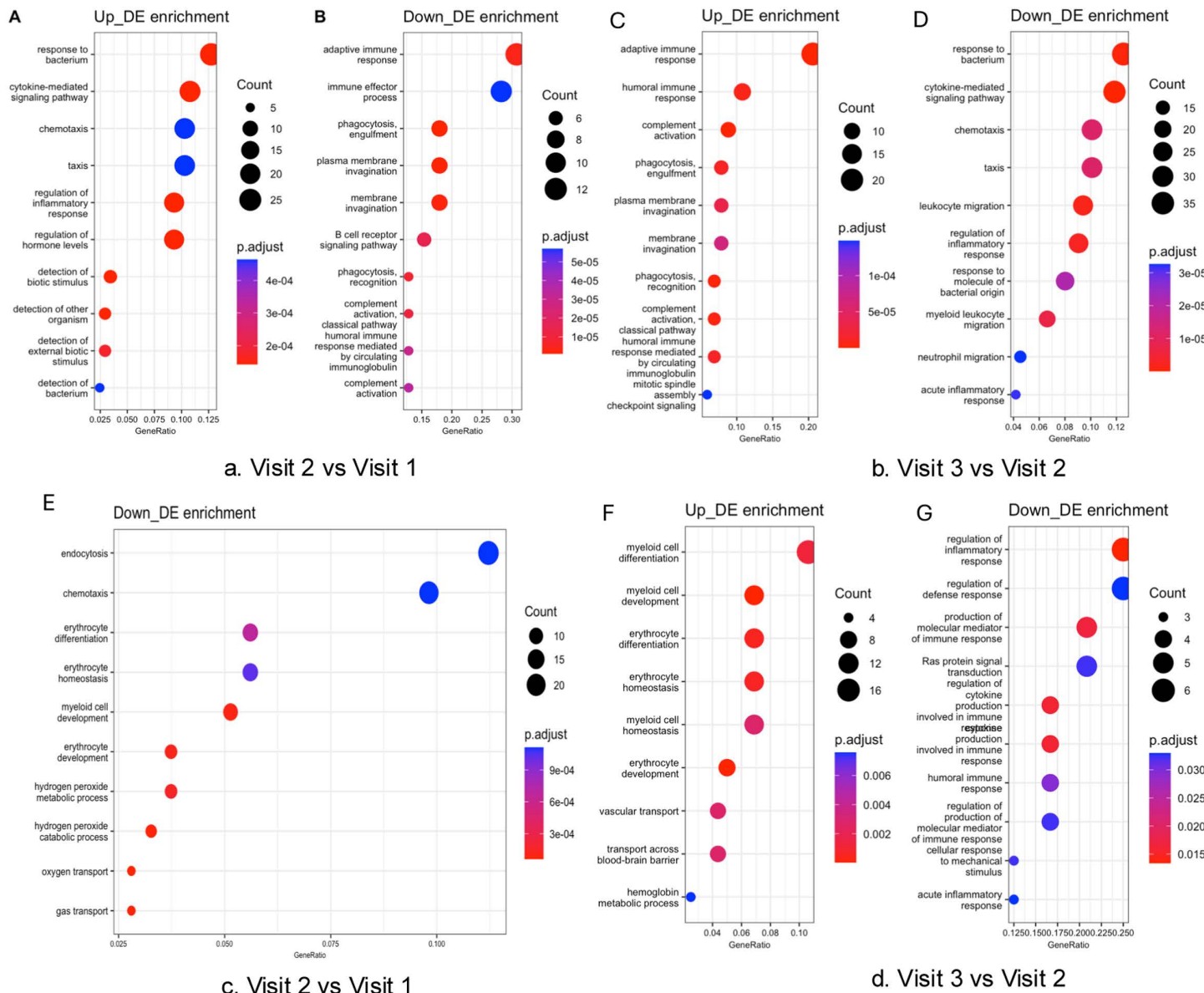

**Fig 5. The top 10 significantly enriched biological processes in pain trajectory 1 and trajectory 2 over time.** Figs 5a and 5b presented the pain trajectory 1 enriched biological processes over time. Figs 5c and 5d presented the pain trajectory 2 enriched biological processes over time.

levels increased at visit 2), had a more pronounced immune-inflammatory response than Pain Trajectory 2 throughout the chemotherapy cycle (Fig 5). Key regulated pathways observed in Pain Trajectory 1 included adaptive immune response, cytokine-mediated signaling pathways, and responses to bacteria. Fatigue Trajectory 1 (fatigue levels increased at visit 2) exhibited an upregulated adaptive immune response and a downregulated bacterial response at visit 3 (Figs 7a and 7b). In contrast, Fatigue Trajectory 2 showed an inflammatory response at visit 2, followed by significant downregulation at visit 3, alongside the upregulation of oxygen-related pathways at visit 3 (Figs 7c and 7d).

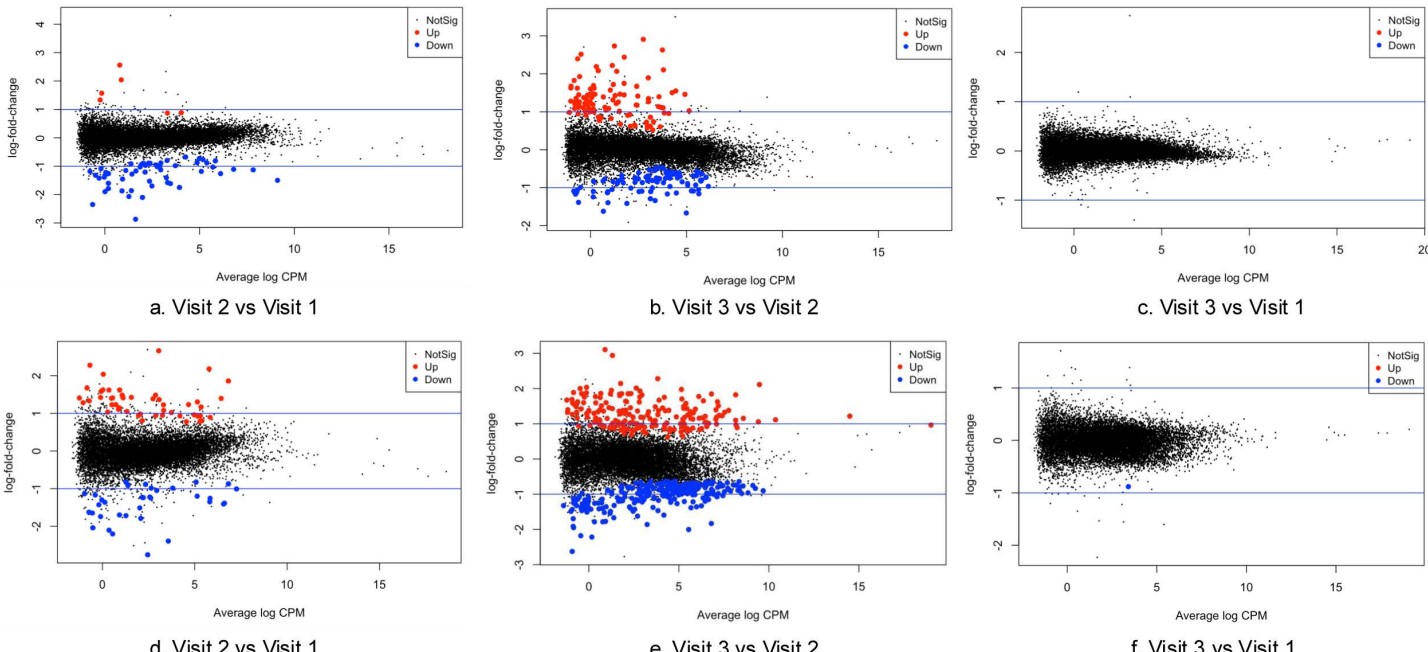

**Fig 6. Mean difference (MD) plot in fatigue trajectory groups over time.** The red dots indicate significantly upregulated genes, while the blue dots indicate downregulated genes. Figs 6a, 6b, and 6c presented the gene differential expression of fatigue trajectory 1 over time. Figs 6d, 6e, and 6f presented the gene differential expression of fatigue trajectory 2 over time.

### Pain, fatigue- and differentially expressed genes

In the LMM analysis (Table 3), patients receiving CAPEOX may experience higher pain severity levels ($\beta = 1.177$, $p = 0.042$). The upregulation of *LILRA6* was associated with higher pain interference ($\beta = -6.621$, $p = 0.010$) and higher fatigue levels ($\beta = -6.621$, $p = 0.010$). Additionally, the downregulation of *CACNG6* ($\beta = -1.043$, $p = 0.047$) and the upregulation of *PRSS33* ($\beta = 1.384$, $p = 0.038$) were linked to increased pain interference. However, given the small sample size, these findings should be interpreted with caution, as they may be subject to Type I error.

### Discussion

This study demonstrated the dynamic profiles of pain, fatigue, and gene expression throughout one chemotherapy cycle and their potential associations. Patients experienced worsening self-reported fatigue and pain patterns after chemotherapy administration. The primary biological perturbations were related to inflammatory responses and myeloid cell development. Despite the limited sample size, findings provided preliminary hypotheses for future research validation, including the potential linkage between elevated pain and fatigue burden, immune-inflammatory responses, erythrocyte functions, and signaling biomarkers (*LILRA6.1*, *CACNG6*, and *PRSS33)*. These insights could lead to more targeted therapeutic interventions to help reduce pain and fatigue in patients with CRC receiving chemotherapy.

The temporal alignment of symptom trajectories with gene expression shifts points to chemotherapy CTX-driven transcriptional reprogramming, particularly those involved in acute immune-inflammatory regulation. The gene expression profile revealed an upregulation of acute inflammatory responses immediately after CTX administration, followed by a rapid downregulation during recovery. This finding aligns with previous research on patients with head and neck cancer undergoing intensity-modulated radiotherapy, which identified significant associations between psychoneurological symptoms, such as depression, fatigue, sleep disturbances, pain, and cognitive dysfunction, with enhanced immune and

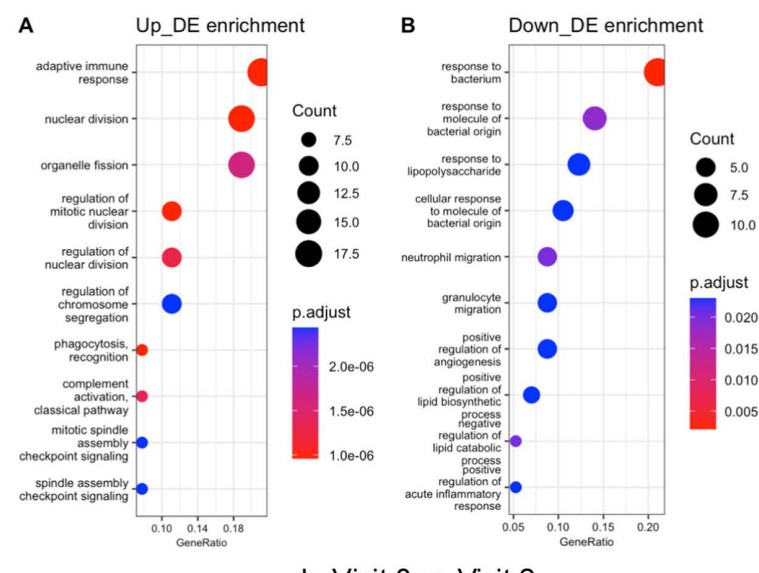

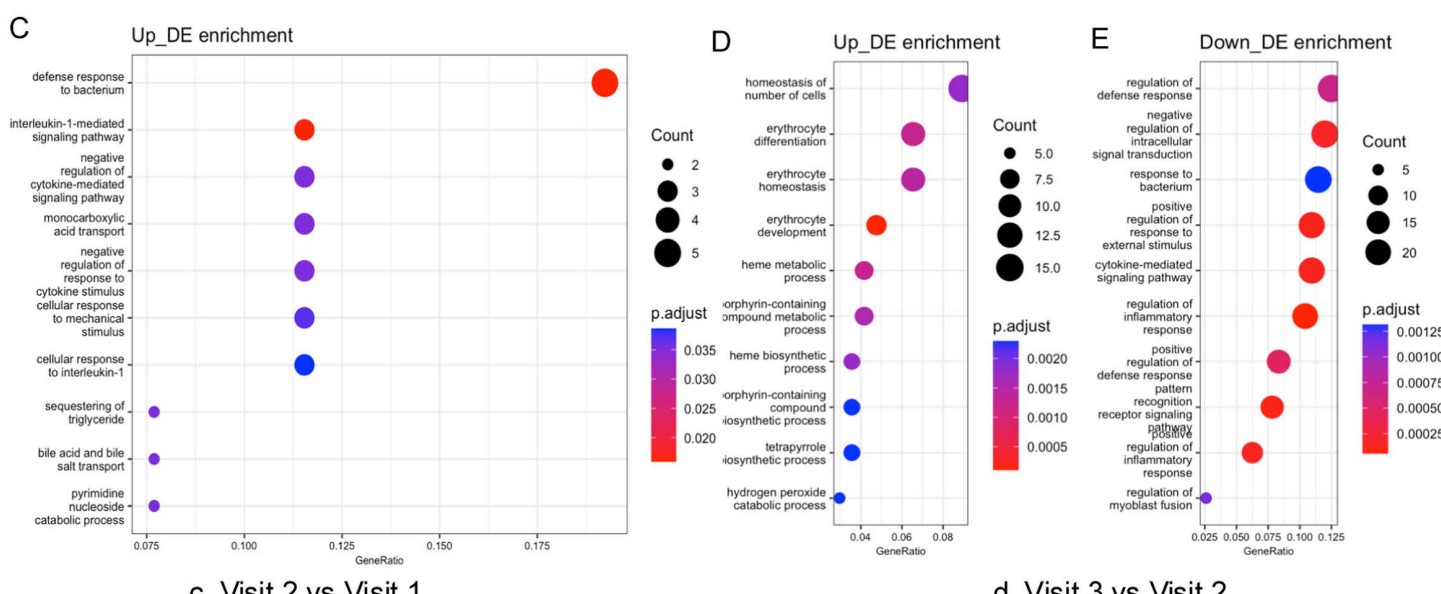

**Fig 7. The top 10 significantly enriched biological processes in fatigue trajectory 1 and trajectory 2 over time.** Figs 7a and 7b presented the fatigue trajectory 1 enriched biological processes over time. Figs 7c and 7d presented the fatigue trajectory 2 enriched biological processes over time.

inflammatory response pathways [31]. In another study of 717 oncology patients, including 15.1% with gastrointestinal cancer, RNA sequencing and microarray analysis were conducted over two CTX cycles to examine the pain-associated pathways. The study found that perturbations in neuroinflammatory pathways were significantly linked to severe pain [40]. Previous studies have demonstrated that acute inflammation triggered by cancer treatments can enhance anti-tumor immunity by promoting dendritic cell maturation, antigen presentation, and effector T cell activation, resulting in more robust anti-tumor responses [41]. However, monitoring inflammatory levels is crucial, as chronic inflammation poses a risk of tumor progression and treatment resistance [41].

**Table 3. The associations between pain/fatigue and genes over time: Linear mixed models (N = 83).**

| Model 1: Pain severity | | | Model 2: Pain interference | | | Model 3: Fatigue | | |
|---|---|---|---|---|---|---|---|---|
| Fixed effect | β | P | | β | P | | β | P |
| (Intercept) | 1.171 | 0.003** | (Intercept) | 2.084 | 0.000** | (Intercept) | 36.158 | <2e-16*** |
| Visit 2 | 0.883 | 0.109 | Visit 2 | −0.330 | 0.554 | Visit 2 | −2.482 | 0.275 |
| Visit 3 | 0.375 | 0.218 | Visit 3 | 0.463 | 0.190 | Visit 3 | −1.841 | 0.192 |
| CAPEOX | 1.177 | 0.042* | CAPEOX | 0.648 | 0.436 | CAPEOX | −2.922 | 0.441 |
| 1st CTX | −0.880 | 0.171 | 1st CTX | −0.288 | 0.746 | 1st CTX | 4.070 | 0.319 |
| LILRA6 | 0.213 | 0.411 | LILRA6 | 1.004 | 0.010* | LILRA6 | −3.492 | 0.033* |
| CDK1 | 0.067 | 0.782 | SLC4A3 | −0.100 | 0.763 | IGHA1 | 0.641 | 0.675 |
| BIRC5 | −0.229 | 0.498 | DNASE1L3 | −0.493 | 0.150 | IGHG1 | −0.561 | 0.708 |
| FFAR3 | −0.184 | 0.474 | CACNG6 | −1.043 | 0.047* | IGLC3 | −3.150 | 0.097 |
| DAAM2 | 0.079 | 0.767 | CDK1 | −0.148 | 0.532 | IGLV144 | 0.338 | 0.773 |
| ADAMTS2 | 0.410 | 0.120 | IL5RA | 0.019 | 0.970 | LYPD2 | 1.519 | 0.236 |
| SYN2 | −0.104 | 0.675 | JCHAIN | 0.623 | 0.244 | PLD4 | 1.439 | 0.496 |
| TOP2A | 0.443 | 0.291 | PRSS33 | 1.384 | 0.038* | SLC4A3 | 0.993 | 0.479 |
| ZDHHC19 | −0.040 | 0.876 | SIGLEC8 | −0.801 | 0.181 | DNASE1L3 | 0.439 | 0.753 |
| | / | / | TNFRSF17 | −0.575 | 0.218 | | | |
| Random effect | Variance | Std.Dev | | Variance | Std.Dev | | Variance | Std.Dev |
| Patient_ID | 1.844 | 1.358 | | 3.806 | 1.951 | | 83.72 | 9.150 |

Note. Signif. codes: 0 '***' 0.001 '**' 0.01 '*' 0.05. CTX. Chemotherapy

Abdominal pain was the most commonly reported site in our study. This may be attributed to hypersensitivity related to the tumor location, digestion, food and gas movement, or epithelial cell damage [17,18]. Pain in CRC may also stem from both local and systemic inflammatory responses triggered by acute chemotherapy-induced injury and cellular senescence. Patients with CRC often exhibited signs of hyperalgesia and central sensitization even at the time of diagnosis, and subsequent treatments may exacerbate these conditions [42]. Jung et al. reported that oxaliplatin injection in rats induced the immediate release of proinflammatory cytokines (IL-1β and TNF-α) in the spinal cord, revealing the molecular pathways in cold and mechanical allodynia [43]. The inflammatory processes may increase the excitability and sensitivity of neurons at the peripheral and spinal levels, potentially leading to pain hypersensitivity [44,45]. However, due to the small sample size, subgroup analyses of abdominal pain mechanisms were not performed. Larger, more diverse cohorts are needed to validate these findings and elucidate pain sensitivity in greater depth.

Our LMMs analyses indicate a potential association between increased *LILRA6* expression and heightened pain interference and fatigue. *LILRA6*, part of the Leukocyte Immunoglobulin-Like Receptor A (*LILRA*) family, is predominantly expressed in immune cells and plays a crucial role in the inflammatory response [46]. Upregulated expression of LILRA6 has been observed in patients with Multiple Sclerosis [47], severe aplastic anemia [48], rheumatoid arthritis [49], and non-small cell lung cancer following neoadjuvant chemo-immunotherapy [50]. Increased LILRA6 can enhance immune cell activation, producing pro-inflammatory cytokines such as Tumor Necrosis Factor Alpha, Interferon-Gamma, and Interleukin-17 [51–53], providing an indirect but plausible link to our findings.

We also found associations between the downregulation of *CACNG6*, the upregulation of *PRSS33*, and elevated pain interference in this study. The *CACNG6* is a gene encoding a subunit of the voltage-dependent calcium channel complex [54]. It is a member of the transmembrane α-amino-3-hydroxy-5-methyl-4-isoxazolepropionic acid (AMPA) receptor regulatory protein (*TARP*) family, which primarily regulates AMPA receptor trafficking and synaptic signaling. Although the role

of *CACNG6* is poorly understood, other members of the TARP family, such as *CACNG2,* have been linked to chronic pain conditions, including post-mastectomy pain in breast cancer patients [55] and neuropathic pain [56,57]. The association between CACNG6 downregulation and pain interference may reflect its role in voltage-gated and ligand-gated ion channels [16]. Reduced *CACNG6* expression may impair calcium channel function, leading to dysregulated pain signaling, heightened neuronal excitability, or a lowered pain threshold [16].

Similarly, *PRSS33* encodes a serine protease predominantly expressed in eosinophils [58]. While the role of PRSS33 in pain has not been previously reported, it is noteworthy that proteases can activate receptor 2 (PAR2) signaling, a mechanism that may exacerbate cancer pain [59]. Furthermore, the involvement of PRSS33 in amplifying inflammatory responses [58] implies a potential correlation with cancer-related pain. Despite these mechanistic insights, our study is the first to implicate LILRA6, CACNG6, and PRSS33 in cancer-related symptoms. Rigorous validation is needed to confirm the causal roles, including the useof preclinical models, such as knockdown and overexpression techniques in cancer-bearing animals to investigate changes in pain and fatigue, as well as replication in clinical cohorts through larger trials with sequential symptom assessments.

Fatigue is a complex, multifactorial symptom in cancer patients, often influenced by inflammation, anemia, neurotransmitter imbalances, and energy metabolism. A previous study found similar regulatory pathways likely contributed to the severity of evening fatigue in breast cancer patients undergoing chemotherapy [60]. Xiao et al. also highlighted that inflammatory markers, including C-reactive protein and interleukin-6 (IL-6), mediated the association between epigenetic age acceleration and fatigue [61]. Other studies revealed that the elevated fatigue levels in CRC patients were associated with the upregulation of B lymphocytes and CD8-positive T lymphocytes, as well as increased transcription factors involved in lymphocyte activation and inflammation (NF-κB, STAT, CREB/ATF, TNF-R1, and IL-6), but reduced activity of interferon regulatory factors (IRFs) [21,62]. Furthermore, the observed down-regulation of myeloid and erythrocyte cell development pathways may be attributed to chemotherapy's nonspecific attack on the normal function of blood cells [63]. This impact can reduce the oxygen-carrying capacity of the blood and may contribute to anemia, which is often associated with fatigue. A rat model study found that 5-fluorouracil had a dose effect on fatigue, cytokines, and markers of anemia [64]. However, it is noteworthy that our study did not observe a significant reduction in hemoglobin levels or white blood cell counts. One possible explanation is that the downregulation in these pathways may not have reached a threshold sufficient to impact cell counts noticeably. Additionally, the body may have engaged compensatory processes to offset the decreased activity in myeloid and erythrocyte pathways, maintaining blood cell production at baseline levels. While the specific role of *LILRA6* in fatigue remains underexplored, its close relationship with other LILRA family members [46], such as LILRA3, may provide insight. Previous studies have shown that increased expression of LILRA3 is positively correlated with disease progression, activity, and treatment response in patients with severe aplastic anemia, a condition associated with bone marrow failure and fatigue [48]. Further validation studies will be imperative to understand the intricate mechanisms linking gene inflammatory responses, erythrocyte functions, and chemotherapy-related fatigue.

Although pain and fatigue frequently co-occur in patients with CRC undergoing chemotherapy, our findings suggested an intriguing difference in the timing of symptom onset between the two subgroups. Previous research has shown that in patients with head and neck cancer undergoing radiotherapy, pain and fatigue often clustered as the most common psychoneurological symptoms, closely associated with upregulated immune-inflammatory responses [31]. In the current study, patients experiencing worsening pain exhibited a more pronounced inflammatory and immune response immediately following chemotherapy. In contrast, those with increased fatigue demonstrated enriched immune-inflammatory activity toward the end of the treatment cycle, accompanied by the downregulation of myeloid and erythrocyte cell development. This temporal divergence highlights the potential for distinct underlying biological mechanisms driving pain and fatigue under chemotherapy, emphasizing the importance of tailored symptom management approaches.

## Strength & Limitations

The relatively small sample size and limited demographic diversity inherent to the pilot study design may limit the generalizability of our findings. Despite the advantages of longitudinal studies in tracking dynamic phenotypes and genotypes, further studies are needed to validate these findings in larger, more racially and ethnically diverse cohorts. The classification of pain and fatigue trajectories highlights two distinct patterns: trajectory 1, where symptoms worsen after chemotherapy (visit 2), and trajectory 2, where symptoms remain unchanged or improve at visit 2. This binary distinction allows a valuable comparison and an initial exploration of the links between symptom changes and biological gene expressions in a limited sample. However, we acknowledge that this method may not fully capture the heterogeneity of symptom progression. Future studies with larger cohorts should apply data-driven approaches, such as clustering or latent class modeling, for more robust trajectory identification. Finally, this study highlights the potential utility of RNA seq as a more precise and objective tool for measuring patient symptoms in future biobehavioral studies.

## Supporting information

**S1 Fig 1. The RNA seq analysis workflow.** The RNA sequencing differential expression (DE) analysis comprised four main stages: preprocessing, mapping, post-processing, and data analysis. During preprocessing, the data underwent quality assessment pre- and post-trimming using *fastqc* and *multiqc,* as well as trimming using *fastp.* In the mapping stage, we selected the human genome hg19 as the mapping index and aligned the reads to the reference genome with *HISTA*. For post-processing, the program HTseq-count was used to count the RNA fragments (i.e., read pairs) mapped to each annotated gene in the genome. Once the counts were generated, we applied R package *edgeR* to perform gene DE analysis. Enrichment analysis was performed using the GOseq packages. Mean-difference (MD) plots and enrichment dot plots illustrated the results for visualization by clusterProfiler package.
(PDF)

**S1 Table 1. Differentially expressed gene list (logFC > 1.5 & FDR < 0.05).**
(PDF)

**S1 Fig 2. Individual symptom trajectories and mean scores among distinct pain/fatigue trajectories.** Individuals who reported higher pain scores at visit 2 compared to visit 1 were classified as belonging to Pain Trajectory 1, while those who reported lower pain scores were classified as Pain Trajectory 2. In a similar way, individuals who experienced increased fatigue levels at visit 2 were categorized as being in Fatigue Trajectory 1, whereas those with decreased fatigue levels were placed in Fatigue Trajectory 2.
(PDF)

## Author contributions

**Conceptualization:** Weizi Wu, Vijender Singh, Andrew Salner, Michelle P. Judge, Xiaomei Cong, Wanli Xu.

**Data curation:** Weizi Wu, Aolan Li, Ming-Hui Chen, Wanli Xu.

**Formal analysis:** Weizi Wu, Vijender Singh, Ming-Hui Chen, Wanli Xu.

**Funding acquisition:** Andrew Salner, Wanli Xu.

**Investigation:** Weizi Wu, Andrew Salner, Wanli Xu.

**Methodology:** Weizi Wu, Aolan Li, Ming-Hui Chen, Michelle P. Judge, Wanli Xu.

**Project administration:** Wanli Xu.

**Supervision:** Weizi Wu, Michelle P. Judge, Xiaomei Cong, Wanli Xu.

**Validation:** Weizi Wu, Aolan Li, Vijender Singh, Andrew Salner, Ming-Hui Chen, Michelle P. Judge, Xiaomei Cong, Wanli Xu.

**Visualization:** Weizi Wu, Aolan Li, Vijender Singh, Andrew Salner, Ming-Hui Chen, Wanli Xu.

**Writing – original draft:** Weizi Wu.

**Writing – review & editing:** Weizi Wu, Aolan Li, Vijender Singh, Andrew Salner, Ming-Hui Chen, Michelle P. Judge, Xiaomei Cong, Wanli Xu.

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
