## [Decision Letter · Decision Letter 0]

Dear Dr. Xu,

Thank you for submitting your manuscript to PLOS ONE. After careful consideration, we feel that it has merit but does not fully meet PLOS ONE’s publication criteria as it currently stands. Therefore, we invite you to submit a revised version of the manuscript that addresses the points raised during the review process.

We look forward to receiving your revised manuscript.

Kind regards,

Keun-Yeong Jeong

Academic Editor

PLOS ONE

Journal Requirements:

“This work was made possible by the Oncology Nursing Foundation, which provided research support through the RE01 grant in 2019. No specific grant number was assigned. Wanli Xu, the corresponding author, received support for this research.”

Reviewers' comments:

Reviewer's Responses to Questions

**Comments to the Author**

1. Is the manuscript technically sound, and do the data support the conclusions?

Reviewer #1: Yes

Reviewer #2: Yes

2. Has the statistical analysis been performed appropriately and rigorously?

Reviewer #1: I Don't Know

Reviewer #2: Yes

3. Have the authors made all data underlying the findings in their manuscript fully available?

Reviewer #1: Yes

Reviewer #2: Yes

4. Is the manuscript presented in an intelligible fashion and written in standard English?

Reviewer #1: No

Reviewer #2: Yes

Reviewer #1: The manuscript titled "Pain, Fatigue, and Associated Gene Expressions Over Chemotherapy in Patients with Colorectal Cancer" presents a compelling and timely investigation into the dynamic changes in pain, fatigue, and associated gene expressions in colorectal cancer patients undergoing chemotherapy. The longitudinal study design is commendable, and the integration of self-reported symptom assessments with RNA sequencing provides valuable insights into the biological underpinnings of chemotherapy-induced symptom burden. The study’s focus on inflammatory pathways and their relationship with pain and fatigue enhances its significance in oncology and symptom management research.

The manuscript is well-structured, and the authors effectively contextualize their findings within the existing literature. The statistical approaches used for differential gene expression analysis and linear mixed models are appropriate for the study design. The discussion section highlights important implications for future research and potential clinical applications. However, there are major concerns about the current version of this manuscript as described below.

Major Concerns:

oThe study includes a relatively small sample size (n=34), which limits the generalizability of the findings. Given the complexity of pain and fatigue mechanisms, a larger cohort is needed to confirm the associations between symptom trajectories and gene expression changes.

oThe demographic homogeneity of the sample (97% white and non-Hispanic) further restricts the applicability of the results to more diverse populations.

oWhile the authors perform appropriate statistical tests, the interpretation of certain results is overly strong given the small sample size. For example, the associations between LILRA6 expression and pain/fatigue (p = 0.010) and between PRSS33 upregulation and pain interference (p = 0.038) should be reported with caution, acknowledging the potential for Type I error.

oThe post-hoc analysis for symptom changes over time lacks a clear adjustment for multiple comparisons. A more conservative approach, such as Bonferroni correction, may be needed.

oThe discussion effectively links the findings to immune-inflammatory pathways, but it lacks a mechanistic explanation of how chemotherapy-induced inflammation translates into altered gene expression patterns. Including references to experimental studies on the molecular pathways of pain and fatigue could strengthen this section.

oThe role of the identified genes (e.g., LILRA6, CACNG6, and PRSS33) in pain and fatigue needs further validation. The authors should discuss whether these genes have been implicated in previous research on cancer-related symptoms.

oFigures 2 and 3 illustrating gene expression changes and pathway enrichment are informative but somewhat difficult to interpret. Providing clearer legends and emphasizing the key findings in the text would improve comprehension.

oThe trajectory classification for pain and fatigue appears somewhat arbitrary. The authors should clarify the criteria used to distinguish between different trajectory groups and consider alternative clustering methods for greater robustness.

Minor Concerns:

oThe manuscript sometimes alternates between ‘chemotherapy’ and ‘CTX.’ It would be helpful to standardize the terminology throughout.

oThe abbreviation ‘LMM’ (Linear Mixed Models) is introduced without a clear explanation. A brief clarification in the methods section would be beneficial.

oThe manuscript states that ‘some restrictions apply’ to data availability. The authors should provide more details on how interested researchers can access the dataset and under what conditions.

Reviewer #2: The manuscripts provides good insight regarding the effect of chemotherapy on patients. However, due to small size sample the result of the research may not be generalizable. Inspite of this, I recommend that the paper is publishable.

**Do you want your identity to be public for this peer review?** For information about this choice, including consent withdrawal, please see our Privacy Policy

Reviewer #1: No

Reviewer #2: **Yes: ** Tamiru Demeke

---

## [Author Response · Author response to Decision Letter 1]

26 Mar 2025

Response to Reviewer #1

Praising the Article

1. The manuscript titled "Pain, Fatigue, and Associated Gene Expressions Over Chemotherapy in Patients with Colorectal Cancer" presents a compelling and timely investigation into the dynamic changes in pain, fatigue, and associated gene expressions in colorectal cancer patients undergoing chemotherapy. The longitudinal study design is commendable, and the integration of self-reported symptom assessments with RNA sequencing provides valuable insights into the biological underpinnings of chemotherapy-induced symptom burden. The study’s focus on inflammatory pathways and their relationship with pain and fatigue enhances its significance in oncology and symptom management research. The manuscript is well-structured, and the authors effectively contextualize their findings within the existing literature. The statistical approaches used for differential gene expression analysis and linear mixed models are appropriate for the study design. The discussion section highlights important implications for future research and potential clinical applications. However, there are major concerns about the current version of this manuscript as described below.

Response: We sincerely appreciate the reviewer’s thoughtful evaluation of our work and the recognition. Regarding the major concerns raised, we have implemented substantial revisions to strengthen methodological transparency, biological plausibility, and clinical interpretability. Key improvements include: 1). enhanced mechanistic explanations linking chemotherapy-induced inflammation to transcriptional changes and symptoms (Responses to Comments 5-6); 2). revised figures with clearer annotations of key pathways and dynamic patterns (Responses to Comment 7); 3). expanded discussion of limitations (e.g., sample size and diversity) and future validation strategies (Responses to Comments 1-4); 4). Validated the trajectory classification (Responses to Comment 8). We believe these revisions address the reviewer’s concerns and below are point-by-point responses to each concern.

Major concerns

1. The study includes a relatively small sample size (n=34), which limits the generalizability of the findings. Given the complexity of pain and fatigue mechanisms, a larger cohort is needed to confirm the associations between symptom trajectories and gene expression changes.

Response: Thank you for your valid concern regarding the sample size. We acknowledged the limit of the sample size and the generalizability of our findings and addressed these limitations in our discussion session (clean version of lines 307-310 on page 18, lines 339-341 on page 19, lines 368-371 on page 20, lines 395-397, 410-414 on page 22). Due to the nature of the longitudinal design, we were able to collect repeated measures to capture the dynamic associations between symptoms and genes. We also applied multiple-testing corrections (FDR <0.05, |logFC| >1.5), which supports the robustness of our key findings. In the updated discussion session, we discussed that validation is needed in fully-powered studies. We look forward to utilizing this preliminary data to secure additional funding support and enhance collaboration with multi-institutional initiatives to expand validation.

2. The demographic homogeneity of the sample (97% white and non-Hispanic) further restricts the applicability of the results to more diverse populations.

Response: Thank you for your valid concern. We agree that the limited demographic diversity may obscure culturally or biologically driven differences in symptom-gene relationships. In the revised discussion, we have explicitly highlighted that the relatively small sample size and limited demographic diversity inherent to the pilot study design may limit the generalizability of our findings, noting the critical need for replication in larger cohorts and racially and ethnically diverse populations. Please refer to the clean version of lines 410-414 on page 22, “Strength & Limitations”.

3. While the authors perform appropriate statistical tests, the interpretation of certain results is overly strong given the small sample size. For example, the associations between LILRA6 expression and pain/fatigue (p = 0.010) and between PRSS33 upregulation and pain interference (p = 0.038) should be reported with caution, acknowledging the potential for Type I error.

Response: We appreciate your insightful observation and reminder to ensure the rigor of the manuscript. The manuscript has been revised to clearly caution against the overinterpretation of these results, specifically in the Abstract (Page 2, Lines 43-44) and Results (Page16, Lines 296-298), and Discussion sections to underscore the exploratory nature of these findings. Furthermore, we have indicated in the discussion that these associations are framed as hypothesis-generating, necessitating validation in larger clinical cohorts or preclinical models. Please refer to the clean version lines 307-310 on page 18, pages 20-21 lines 368-371.

4. The post-hoc analysis for symptom changes over time lacks a clear adjustment for multiple comparisons. A more conservative approach, such as Bonferroni correction, may be needed.

Response: We appreciate your suggestion. In our post-hoc analysis, we implemented the Bonferroni correction for the pairwise comparisons across each outcome variable (e.g., three time-point comparisons per outcome: V2 vs V1, V3 vs V2, V3 vs V1), and the post-hoc P value denotes the corrected P value. For instance, in the total fatigue score, the comparison between V2 vs V1 yielded a significant result after applying the Bonferroni correction (p=0.012), as did the comparison of V3 vs V2 (p=0.018). We have revised the table to clarify that these post-hoc analysis P values represent the corrected P values, and have included a footnote to elucidate the correction method. Please refer to the clean version of lines 218-222 on page 12, “Table 2. Repeated Measure ANOVA results for main outcomes (N=34).”

5. The discussion effectively links the findings to immune-inflammatory pathways, but it lacks a mechanistic explanation of how chemotherapy-induced inflammation translates into altered gene expression patterns. Including references to experimental studies on the molecular pathways of pain and fatigue could strengthen this section.

Response: We sincerely thank the reviewer for this insightful suggestion. To address the need for clearer mechanistic links between chemotherapy-induced inflammation and transcriptional changes, we have substantially revised the Discussion section as follows: 1). Added molecular pathways bridging inflammation and chemotherapy and symptoms: We now describe how chemotherapy agents using a rat model prompted the immediate release of proinflammatory cytokines (IL-1β and TNF-α) in the spinal cord, revealing the molecular pathways in chemotherapy-related pain development (Jung et al., 2017). For fatigue, we discuss that the elevated fatigue levels in CRC patients were associated with the upregulation of B lymphocytes and CD8-positive T lymphocytes, as well as increased transcription factors involved in lymphocyte activation and inflammation (NF-κB, STAT, CREB/ATF, TNF-R1, and IL-6), but reduced activity of interferon regulatory factors (IRFs) (Black et al., 2018; Wang et al., 2012). 2). Benchmarked against analogous multi-omics studies: While our cohort focused on colorectal cancer, we now compare our gene expression patterns to similar transcriptomic analyses in gastrointestinal cancer (Shin et al., 2023) and head and neck cancer (Lin et al., 2023). Despite population differences, these studies consistently implicate inflammation dysregulation in chemotherapy-related symptoms, reinforcing the generalizability of our mechanistic framework. These revisions provide a tighter conceptual link between chemotherapy’s inflammatory effects, transcriptional changes, and symptom biology. Please refer to the highlighted sections in yellow in the clean version in the Discussion section (lines 316-323 on page 18, lines 334-337 on page 19, lines 377-381 on page 21).

6. The role of the identified genes (e.g., LILRA6, CACNG6, and PRSS33) in pain and fatigue needs further validation. The authors should discuss whether these genes have been implicated in previous research on cancer-related symptoms.

Response: Thanks to the reviewer for highlighting this critical point. While our Discussion section references previous studies concerning LILRA6, CACNG6, and PRSS33 in the contexts of inflammation, we have now emphasized more mechanistic interpretation between the established functions of these genes and their potential biological associations with cancer symptoms (for instance, LILRA6 → pro-inflammatory cytokines → pain/fatigue); incorporated citations that link potential pathways (for example, the regulation of voltage-gated and ligand-gated ion channels via CACNG6 (Wistrom et al., 2022) and the activation of the protease receptor 2 (PAR2) signaling mechanism (Lam & Schmidt, 2010) related to pain) to cancer-related pain; and explicitly acknowledged that this is the first report to emphasize the necessity for further validation within preclinical models or cancer clinical cohorts. Please refer to lines 343-345 on page 19 and lines 356-371 on pages 20-21 in the clean version.

7. Figures 2 and 3 illustrating gene expression changes and pathway enrichment are informative but somewhat difficult to interpret. Providing clearer legends and emphasizing the key findings in the text would improve comprehension.

Response: We appreciate the reviewer for the valuable suggestion. We have revised the legends for Figures 2 and 3 to provide additional context for interpreting the figures. Additionally, we have summarized the key enriched pathways and findings in the text. Please refer to lines 238-254 on pages 13 and 14 in the clean version.

8. The trajectory classification for pain and fatigue appears somewhat arbitrary. The authors should clarify the criteria used to distinguish between different trajectory groups and consider alternative clustering methods for greater robustness.

Response: Thank you for your insightful comment. We recognize that the description in the limitations section may have been misleading. To address this, we have revised the limitations section and provided a clearer explanation of the classification criteria in the Results section. After careful consideration and discussions with our research team, including our statistician, we confirm that our trajectory group classifications are designed to be as objective as possible. We agree that more robust trajectory classification methods (e.g., clustering or latent class growth modeling) could enhance trajectory identification. However, due to the limited sample size, such data-driven approaches would likely yield unstable and non-generalizable clusters. For this exploratory analysis, we therefore opted for a clinically interpretable, binary classification based on symptom change direction. We now clarify this rationale in the Results and acknowledge the limitation in the strength and limitation section. Please refer to the clean version of lines 259-263 on page 14, and lines 41-421 on pages 22-23.

Minor concerns

1. The manuscript sometimes alternates between ‘chemotherapy’ and ‘CTX.’ It would be helpful to standardize the terminology throughout.

Response: We thank the reviewer for noting this inconsistency. We have standardized the term to "chemotherapy" throughout the manuscript to improve clarity and avoid confusion.

2. The abbreviation ‘LMM’ (Linear Mixed Models) is introduced without a clear explanation. A brief clarification in the methods section would be beneficial.

Response: Thanks for the considerable suggestion. We now briefly explain its suitability for analyzing longitudinal data with repeated measures, as well as its fix and random effect. Please refer to the clean version of lines 175-180 on page 9, under “Pain/fatigue trajectories related to differentially expressed genes.”

3. The manuscript states that ‘some restrictions apply’ to data availability. The authors should provide more details on how interested researchers can access the dataset and under what conditions.

Response: Thanks for prompting us to clarify data accessibility. A detailed Data Availability Statement has been added to the end of the manuscript (Lines 434-444 on pages 23-24) with the following provisions: 1). De-identified clinical and transcriptomic data are available upon request to the corresponding author, subject to a signed Data Use Agreement ensuring ethical compliance. 2). Analysis code and preprocessing pipelines can be accessed by github link under first author’s account. These measures ensure transparency while protecting participant confidentiality per institutional IRB guidelines.

Response to Reviewer #2

1. The manuscripts provides good insight regarding the effect of chemotherapy on patients. However, due to small size sample the result of the research may not be generalizable. Inspite of this, I recommend that the paper is publishable.

Response: We sincerely appreciate your constructive feedback and endorsement of our work. We fully agree that the small sample size limits generalizability, a limitation we now emphasize more explicitly in the revised Discussion (clean version of lines 307-310 on page 18, lines 339-341 on page 19, lines 368-371 on page 20, lines 395-397, 410-414 on page 22). We have also indicated in the discussion that these associations are framed as hypothesis-generating, necessitating validation in larger clinical cohorts or preclinical models. Thank you for recognizing the value of this work despite its constraints. We have revised the manuscript and look forward to utilizing this preliminary data to secure additional funding support and enhance collaboration with multi-institutional initiatives to expand validation.

Response to Editors

Response: Thanks. We have followed the rules as instructed.

2. Please provide an amended statement that declares *all* the funding or sources of support (whether external or internal to your organization) received during this study, as detailed online in our guide for authors at http://journals.plos.org/plosone/s/submit-now. Please also include the statement “There was no additional external funding received for this study.” in your updated Funding Statement. Please include your amended Funding Statement within your cover letter. We will change the online submission form on your behalf.

Response: Thank you for your reminders. We have included all funding sources and added the statement, "There was no additional external funding received for this study," in the revised manuscript and the cover letter for your reference. I appreciate your assistance in updating this on your end.

b) If there are no restrictions, please upload the minimal anonymized data set necessary to replicate your study findings to a stable, public repository and provide us with t

---

## [Decision Letter · Decision Letter 1]

Pain, fatigue, and associated gene expressions over chemotherapy in patients with colorectal cancer

PONE-D-25-02816R1

Dear Dr. Xu,

We’re pleased to inform you that your manuscript has been judged scientifically suitable for publication and will be formally accepted for publication once it meets all outstanding technical requirements.

Kind regards,

Keun-Yeong Jeong

Academic Editor

PLOS ONE

Additional Editor Comments (optional):

Reviewers' comments:

Reviewer's Responses to Questions

**Comments to the Author**

Reviewer #2: All comments have been addressed

2. Is the manuscript technically sound, and do the data support the conclusions?

Reviewer #2: Yes

3. Has the statistical analysis been performed appropriately and rigorously?

Reviewer #2: Yes

4. Have the authors made all data underlying the findings in their manuscript fully available?

Reviewer #2: Yes

5. Is the manuscript presented in an intelligible fashion and written in standard English?

Reviewer #2: Yes

Reviewer #2: The authors addressed all the comments forwarded to them sufficiently. Therefore, in my opinion, the paper can be published.

**Do you want your identity to be public for this peer review?** For information about this choice, including consent withdrawal, please see our Privacy Policy

Reviewer #2: **Yes: ** Tamiru Demeke

---

## [Editor Report · Acceptance letter]

PONE-D-25-02816R1

PLOS ONE

Dear Dr. Xu,

I'm pleased to inform you that your manuscript has been deemed suitable for publication in PLOS ONE. Congratulations! Your manuscript is now being handed over to our production team.

Kind regards,

on behalf of

Dr. Keun-Yeong Jeong

Academic Editor

PLOS ONE